# Using Food Models to Enhance Sugar Literacy among Older Adolescents: Evaluation of a Brief Experiential Nutrition Education Intervention

**DOI:** 10.3390/nu11081763

**Published:** 2019-07-31

**Authors:** María Isabel Santaló, Sandra Gibbons, Patti-Jean Naylor

**Affiliations:** School of Exercise Science, Physical and Health Education, Faculty of Education, University of Victoria, Victoria, BC V8P 5C2, Canada

**Keywords:** sugar, knowledge, education intervention, food models, adolescent

## Abstract

Adolescent diets high in sugar are a public health concern. Sugar literacy interventions have changed intake but focused more on children, adults, and early adolescents and on sugar sweetened beverages rather than total sugar consumption. Food models are an efficacious experiential learning strategy with children. This study assessed the impact of two 45 min nutrition lessons using food models on adolescents’ sugar literacy. Classes (*n* = 16) were randomized to intervention or control with knowledge, label reading skills, intentions to limit sugar consumption measured at baseline and follow-up. Two hundred and three students aged 14 to 19 from six schools on Vancouver Island, BC, Canada participated in the study. Adolescents’ knowledge of added sugar in foods and beverages and servings per food group in a healthy diet was limited at baseline but improved significantly in the intervention condition (F(1, 201) = 104.84, *p* < 0.001) compared to controls. Intention to consume less added sugar increased significantly after intervention (F(1, 201) = 4.93, *p* = 0.03) as did label reading confidence (F(1, 200) = 14.94, *p* < 0.001). A brief experiential learning intervention using food models was efficacious for changing student’s knowledge about sugar guidelines and sugar in food, label reading confidence, and intention to change sugar consumption.

## 1. Introduction

Sugar in the diet is present naturally or as an added ingredient which has been related to both tooth decay and excess weight gain [1]. Overweight and obesity in populations has been rising worldwide, and is a significant public health concern as they are associated with health problems such as diabetes and cardiovascular disease [2]. Canadian adolescent overweight and obesity rates have mirrored this pattern; with 30% of children age 5 to 17 overweight or obese in 2018 [3].

It is well accepted that a major cause of overweight and obesity in the last three decades has been sedentary lifestyles and changes in the diet including consumption of less fruits and vegetables and more energy dense foods and beverages in the diet [4]. One World Health Organization (WHO) recommendation for addressing the overweight and obesity epidemic is limiting intake from total fats and free sugar, the sugar added to food and beverages, and sugars naturally present in honey, syrups, and fruit juices [5].

Although there is not yet an international consensus around the maximum amount of sugar that should be present in a healthy diet, recommendations from the WHO are widely used. These state that less than 10% of the calories in the diet should come from free sugar [5]. In Canada, adolescents are consuming 14.1% of their calories from added sugar [6] (sugars that are incorporated into foods and beverages and sugars naturally present in honey, syrups, and undiluted juices concentrates [7]). Added sugar is consumed by adolescents in sugar-sweetened beverages (SSBs) but in solid foods as well [8]. Adolescent sugar consumption is a concern since this is an age where eating habits are formed and diet quality deteriorates [9]. Strategies and interventions to change this public health trend are needed. One approach has been to address food literacy, which is not only about knowledge and awareness, but also concerns the skills/capacity to act [10].

Brooks and Begley [11] reviewed literature focused on adolescent food literacy and found there was a lack of food literacy interventions for older adolescents (i.e., high school students). Additionally, based on their review of effective interventions, they suggested using innovative teaching aids, including opportunities for experiential or “hands-on” learning. Experiential learning is designed to develop personal understanding, knowledge, skills, and attitudes through active engagement and reflection on certain activities [12]. Atkins and Michie [13] suggest that it is important to include one or more behavioral change techniques to influence capacity building, motivation or opportunity to change behavior (COM-B model). Finally Brooks and Begley suggested schools, community centers or sporting clubs as possible delivery settings for food literacy intervention.

Schools represent an ideal setting to facilitate dietary behavior change since youth regularly attend for prolonged periods across a year and as they age [14] and they are responsible for the delivery of health curricula. Therefore, the school presents an opportunity to provide nutrition-related education interventions that incorporate evidence-based intervention techniques to increase health literacy levels.

In British Columbia, Canada, the Physical and Health Education (PHE) curriculum for secondary students includes teaching students about the role of nutrition in health and performance in order to help them develop the ability to choose to eat healthy foods [15,16]. The Ministry of Education encourages flexibility in finding different ways to bring this learning to students. Including sugar literacy education sessions as part of physical and health education could be a way to help adolescents make healthy food choices.

Nutrition education addressing sugar literacy for adolescents has mostly targeted SSB consumption [17,18]. Sugar-sweetened beverages are the single category of food/beverages through which adolescents consume the most added sugar; although the highest amount of added sugar they consume comes from across all other food groups together [6]. Therefore, interventions that consider not only knowledge, awareness, and skills related to sugar in SSBs but also in other foods are vital to reducing total added sugar in the diet. There appears to be no published literature related to broader sugar literacy interventions targeting older adolescents although a recent study intervened with 10 to 12 year old children in the United Kingdom [19]. The researchers provided two 45 min educational sessions across 2 days that also incorporated experiential activities using a teaching aid (heaped teaspoons of sugar to illustrate the amount of sugar in a variety of foods). At baseline, children had limited knowledge of sugar in foods and beverages. The educational intervention improved their knowledge significantly at follow-up (*p* < 0.001). This is not surprising, as visual teaching aids have been shown to make learning more effective. Shabiralyani et al. [20] suggest that 83% of what is learned is gained from the sense of sight, 11% from what we hear, and the remaining from the senses of smell, touch, and taste.

Life-sized food models are visual aids that have been used in nutrition education. The literature shows that two- or three-dimensional food models have been used to assess the amount of food consumed [21,22,23,24,25], to examine and teach nutrition knowledge in early childhood [26,27,28] and for nutrition education with families [29]. However, to date, there appears to be little evidence about the use of life-sized food models for nutrition education with adolescents. Two-dimensional food models may be a more feasible option for use in schools because of their lower cost, weight, and size but their efficacy needs to be tested.

The purpose of this study was to explore the impact of two 45 min nutrition lessons using life-sized two-dimensional food models on adolescents’ sugar literacy. Specifically, the study objectives were to determine the effect of this experiential learning strategy on the knowledge and awareness of sugar content in foods and beverages and the recommendations for limits on sugar consumption as well as the impact on skills and intention to consume less added sugar.

## 2. Materials and Methods

The research design included a randomized controlled trial with baseline and follow-up measures (see Figure 1), with randomization by class to either a regular Physical and Health Education class condition (control group) with delayed nutrition education sessions or two nutrition education sessions using food models during physical and health education (intervention group).

The sample size was calculated using G*Power analysis program [30] with the power set at 0.9, significance at *p* < 0.05, and an effect size d = 0.8 (large effect). A minimum of forty-nine participants was needed in each condition to be able to detect a difference between groups.

A 5 min recruitment presentation explaining the project was shared at a Physical and Health Education teachers meeting (representing 5 different school districts on Vancouver Island). Teachers were asked to contact the researcher if they were interested.

Eleven teachers from six schools in six communities indicated interest in participating in the project. An information letter and a consent to participate form were sent to the principals of the six schools. After University Human Research Ethics approval (#17-477), principal consent, and school district approval were received, a presentation about the study (i.e., objectives, participation dates, and evaluation guidelines) was conducted with the participating teachers’ PHE classes, and an information package including consent forms asking for their and their parents’ consent to participate in the research component (measurement) was handed out to each student. The schedule for nutrition education sessions was coordinated directly with each teacher.

Sixteen classes with a total of 334 students received the two 45 min nutrition education sessions. Each school participated with either 2 or 4 classes taking part. Half of the classes in each school were randomly assigned to the usual practice control condition (*n* = 8) and half to intervention conditions (*n* = 8). The consort study flow diagram (see Figure 2) shows that 214 students consented, 110 were in the intervention condition and 104 in the control group. All students that were present received the nutrition education as part of their physical education curriculum; however, only questionnaire data from the students that consented and participated in the baseline measure were analyzed. All control class students received the intervention after measurement was completed.

The primary purpose of the nutrition education sessions was to enhance sugar literacy (knowledge, awareness, skills and intention to change) among youth. The sessions included an experiential learning component according to Michie’s behavior change model (COM-B; [31]) to increase knowledge and understanding around sugar content in food and beverages and the daily amount of added sugar in a healthy diet placed into the context of Canada’s Food Guidelines to promote healthy eating and lower consumption of added sugar. The sessions also included a component to help students develop the necessary skills to interpret nutrition fact labels in packaged products and design healthy eating patterns.

Each nutrition education session took place during PHE class time (45–65 min depending on the school schedule) and was held either in a room with large tables or on the gym floor so students could have enough space to work with the food models. Participants were provided with 2 didactic lectures plus experiential learning activities. For the experiential learning activities, participants were teamed in groups of 3 or 4, and each team was given a kit with more than 150 life-sized food and beverage models, 100 models of sugar teaspoons, and colored copies of seven nutrition fact labels of commercial foods and beverage packages. NutriKitUSA and Sip Smart BC life-sized two-dimensional food and beverage models with nutrition information including added sugar on the back were used (see the example in Appendix A). The food models in the kit included servings of fruits, vegetables, grain products, milk products, protein-rich foods, and beverages. For understanding added sugar content in packaged foods and beverages, seven additional nutrition fact labels of sugar-containing foods and beverages were analyzed.

The schedule and intervention curriculum content are described in detail in the following section.

Session 1:

Lecture: A brief educational lecture focused on overall diet and diet quality, which includes:Canada’s Food Guide and the number of recommended daily servings from each food group for the age of participants;Limiting consumption of added sugar as part of a healthy lifestyle;The use of two-dimensional food models.

Experiential learning activities: food models were used to:Visualizing an everyday personal diet (students used the models to represent their previous day’s diet);Designing a daily menu according to Canada’s Food Guide Servings (students were asked to use the food models to create a healthy daily diet according to food guide servings)Setting goals on how to modify the personal diet to consume the number of servings suggested by Canada’s Food Guide

Session 2:

Lecture: A brief educational lecture focused solely on sugar content in food and beverages, including the following:World Health Organization recommendations/guidelines for consumption of sugars;How to read and interpret ingredients lists on packaged foods and beverages, specifically, the added sugar content in Nutrition Fact tables.

Experiential learning activities: use of two-dimensional food models (including models of teaspoons of sugar) to help students (see Figure 3):Visualizing the amount of added sugar in food and beverages by expressing the content of sugar in teaspoons of sugar;Identifying added sugar information in packaged food (real food labels were provided) and converting added sugar content to teaspoons of sugar;Analyzing an example of an adolescent diet (supplied by the researcher) and determining the teaspoons of added sugar included in that diet. Make recommendations on how to reduce total added sugar and incorporate the right amount of servings from each Food Group for that particular diet.

The following section describes research procedures, measurement instruments and data analysis.

### 2.1. Procedures

After randomization, visits to each class were scheduled. All students had the benefit of the nutrition education lessons; however, intervention groups had the lessons between baseline and follow-up measures and the control group received them after the follow-up measure (see Figure 2). Measures and nutrition education lessons took place April–May 2018. Due to the pragmatic constraints, baseline measures were implemented just prior to the first education session and follow-up measures immediately after session 2. Control condition students completed the baseline measure on the same day as the intervention students but did not receive the two sessions. They received the first session immediately after the follow-up measure, and then a second session was scheduled.

### 2.2. Measurement Instrument

The 20 item survey instrument (completed at both baseline and follow-up) was composed of modified questions extracted purposively from two validated questionnaires: the Canadian Behavior, Attitude, and Nutrition Knowledge Survey [32] and the Intentions to Eat Low-Glycemic Index Foods questionnaire [33] and is described following (see the instrument in Appendix B):

Demographics: The age, gender, and grade of the participants were asked. Each participant was assigned a code to protect the confidentiality of their data.

### 2.3. Experimental Variables

Knowledge of daily portions: the questions were open-ended and asked the student to approximate the number of daily food group servings according to Canada’s Food Guide for their particular age and gender.

Knowledge of the maximum amount of added sugar consumption: this item was open-ended and asked the maximum number of teaspoons of added sugar that an average person (14 years and up) could eat each day to maintain a healthy diet.

Knowledge of added sugar content in food and beverages: four questions for sugar content in different foods and two for different beverages were included.

Self-efficacy: Five items with a 7 point Likert scale were included. Scale reliability analysis showed the self-efficacy measure was internally consistent (Cronbach’s Alpha = 0.74). The five questions were summed together to produce an overall score ranging from 5 to 35 representing their belief in their ability to eat/drink fewer foods and beverages with added sugar.

Intention to eat/drink foods and beverages with less added sugar: Three items that had a 7 point Likert scale with the response format “Strongly disagree to Strongly agree” were included. Cronbach’s alpha for this subscale was 0.839. Items were summed together to produce an overall score. A higher score represented a higher intention to eat/drink fewer foods and beverages with added sugar in the following two weeks.

The frequency of limiting foods high in sugar: a 7 point Likert scale question (from rarely to always) about how often foods high in sugar were limited when making food choices was used.

Nutrition fact label reading confidence. Two 7 point Likert scale questions were used. In the first question, students were asked if, when reading nutrition facts labels, they would look for information about sugar and on the second question, if they were confident that they could interpret that information.

### 2.4. Data Analysis

Data analysis was conducted using SPSS (IBM Corp. Released 2016, Version 24, Armonk, NY, USA). Data were screened for completeness, missing data, and normality. One outlier was eliminated, four outliers in the knowledge variables for the control group were not removed because they were possible values. Twenty-four participants in the control group and 13 in the intervention group completed the baseline measure but were not present for the second measure. Intention-to-treat protocols for missing and absent data were used and the baseline value carried forward. Descriptive statistics (means, SD) were calculated and one-way analysis of variance (ANOVA) was used to determine if there were differences between groups at baseline. A regression analysis determined if there was a relationship between baseline demographic variables, including age and sex and the dependent variables. A repeated measures ANOVA determined if the groups differed over time and by condition. An apriori statistical cut-off point based on an alpha level of 0.05 was used for significance.

## 3. Results

### 3.1. Baseline

Participant characteristics are displayed in detail in Table 1. As noted in Table 1, almost three-quarters of the participants were female. The majority of students in the control group were grade 9 or 10, and for the intervention group, grade 10 or 11 and, thus, the average age of students in the control group was approximately 1 year younger than those in the intervention.

One-way ANOVA scores showed that there was a significant difference in age, sex, and grade by group at baseline. Regression analysis showed, however, that there was no correlation between age (grade) and sex, with the dependent variables at baseline and after the intervention. The percent of variance (*R* squared) explained in the dependent variables by age (grade) and sex was always below 30% (*R*^2^ < 0.3). See Appendix C for regression analysis results.

Baseline means and standard deviations for the dependent variables are also shown in Table 1. This table shows that knowledge was low and that this did not differ by group. No difference was found between groups at baseline for self-efficacy, intention to consume less added sugar, ability to interpret sugar content in food labels, frequency of limiting foods high in added sugar, and frequency of added sugar being read in food labels.

### 3.2. Follow-Up

Means and main effect for time by condition are reported in Table 2 and summarized in the following section. An intervention effect was found for all knowledge variables across time. The intervention condition had a significantly higher score in their knowledge about daily food group portions, the maximum daily recommended amount of added sugar that should be consumed, and the added sugar content in foods and beverages at follow-up compared to controls. Eighty percent of the intervention group correctly answered the maximum recommended amount of added sugar that can be present in a healthy diet compared to 10% of the control group. The intervention group correctly answered 60% of the daily food group portion questions compared to 20% of correct answers for the control group at follow-up. Knowledge about sugar content in foods and beverages increased from 18% to 48% correct answers for the intervention group and was significantly different than the correct responses in the follow-up measure for control group. Effect sizes ranged were large in all knowledge dependent variables.

Intention to consume less food and beverages with added sugar and the ability to interpret sugar content in food labels differed significantly over time as well, with intentions to consume less added sugar increasing in the intervention condition. Effect sizes were respectively small and intermediate (η^2^ = 0.02, η^2^ = 0.07) however. There were no significant differences among groups over time in student’s beliefs in their ability to limit added sugar (self-efficacy) nor in the frequency with which they reported limiting food that contained sugar or read the added sugar on food labels.

## 4. Discussion

With the importance of preventing obesity and enhancing dietary quality during adolescence highlighted in the literature, addressing the adolescent consumption of added sugars is critical. This study addressed a gap in the evidence-based literature by targeting adolescents rather than children and addressing sugar literacy across the spectrum of foods consumed rather than just through sugary drinks. Additionally, it tested a brief nutrition intervention delivered in high school Physical and Health Education classes using food models to provide an experiential learning component. Finally, it incorporated two-dimensional food models which are potentially more feasible to adopt in school-based health promotion efforts.

This study showed that a two 45 min brief nutrition education intervention using food models significantly improved adolescents’ food and sugar literacy. Specifically, the intervention enhanced knowledge of added sugar content in foods and beverages, the maximum amount of added sugar in a healthy diet, food group servings in a healthy diet, and it also increased the ability of adolescents to interpret sugar content in food labels and increased their intention to reduce the consumption of added sugar. Not surprisingly given the post measurement was conducted immediately after the second lesson, the reported frequency of limiting foods with added sugar and reading labels was not significant. These results are discussed in the context of the following literature.

Knowledge has been recognized as an essential component in behavioral change theories of health promotion [34]. Adolescents’ knowledge about added sugar content in food and about recommendations related to the maximum amount of sugar in a healthy diet was low at baseline for all participants, but it increased significantly following the intervention. This is consistent with a previous study that showed that two interactive classroom sessions about sugar given to 10–12 year-old children significantly improved their knowledge of sugar in food and beverages [19]. Further research is needed to evaluate if this knowledge was sustained over time.

This study also examined if adolescents knew the number of servings of each of the food groups that should be present in a healthy diet at their age and if that knowledge changed with the intervention. Even though the study focused on sugar literacy, education about food guidelines was included to ensure students understood healthy eating patterns, how to analyze their own diet, and could place sugar consumption within the context of the overall picture of healthy eating. Adolescents had poor knowledge about servings at baseline, but that knowledge significantly increased with the education sessions; although remaining less than ideal. They used this knowledge to create and visualize a healthy diet utilizing the food models and compared it to their everyday diet. The instrument, however, tested only their knowledge and not their actual ability to use that knowledge to implement a healthy diet. Further studies using eating behavior measurement tools like Food Frequency Questionnaires should test if this type of experiential learning intervention has a short- or long-term impact on the diet.

Having the skills or ability to act is an essential component of food literacy, in this case, sugar literacy. The intervention helped adolescents interpret sugar content in food labels. Nutrition courses including label reading have been given to college students with similar results [35]. Tallant’s [35] study also reported that being able to understand food labels influenced students’ decisions to include healthier foods in their diet. Once again, further studies are needed to evaluate whether or not over time the ability to interpret labels is sustained and if less added sugar is consumed.

This study evaluated if the nutrition education lessons had an effect on intentions to reduce the consumption of added sugar. The intervention group had a higher intention to reduce the amount of added sugar consumed after intervention, but the effect size was minimal. It may be that a brief intervention is less effective for shifting intentions.

Self-efficacy, as the confidence in one’s personal ability to eat/drink less added sugar, was tested but did not change after the intervention. No change was expected after intervention since the follow-up measure was completed immediately after the second nutrition education session. Success is a key component of self-efficacy [36] and they had no opportunity to “apply’” their skills in real-life. Further studies that examine self-efficacy over time are needed.

As suggested by Michie [13], incorporating educational and training techniques did affect knowledge about sugar and increased the skills to interpret sugar content in food, beverages, and food labels. This theoretically influences the capacity (capability) of the students to engage in changing sugar eating behaviors by reducing the amount of added or free sugar consumed. In keeping with the concept of opportunity in Michie’s model, these changes in adolescent sugar consumption could be enhanced by implementing policies to reduce the consumption of foods and beverages high in sugar [37]. Examples of policies suggested by the American Academy of Pediatrics to reduce consumption of SSBs, but that might also be able to be applied to other foods high in sugar as well, are taxes to increase the price of these products and the use of part of the tax revenues to reduce health and socioeconomic disparities, and decreasing marketing to adolescents about products high in added sugar and making healthy low added sugar beverages and foods the default in vending machines, parks, and restaurants [37].

Based on the experience gained in this study, a brief intervention had an impact on important food and sugar literacy variables. Given that session 1 addressed broader food education, an even briefer intervention of one sugar literacy lesson using food models and their impact on awareness of sugar content and sugar consumption in adolescents should be tested. This is consistent with the conclusions of a systematic review on brief nutrition interventions conducted in 2018 that proved that brief interventions can be sufficient to improve short-term dietary behavior [38].

### Limitations

The results of the study should be viewed in the context of the limitations. Validated questionnaire items were adapted and not re-validated. We did re-establish scale reliabilities where more than one item contributed to a sub-scale. We randomized by class not individual student and did not adjust for class as a cluster. We accommodated for school as a cluster by having both an intervention and control class in the same school. This opened the study up to the potential for contamination and a manipulation check question was not included. Finally, due to the pragmatic constraints, actual behavior was not measured. The follow-up measurement was administered immediately after the second nutrition education session, which did not give the participants an opportunity to implement what they learned.

## 5. Conclusions

A brief sugar nutrition education intervention delivered in school Physical and Health Education classes appeared efficacious for increasing sugar literacy among adolescents. Knowledge around the number of portions and sugar content in food and beverages as well as the maximum amount of added sugar in a healthy diet was low in adolescents but increased significantly after just two 45 min nutrition sessions, as did their label reading confidence and intention to consume less added sugar. Increasing sugar literacy in adolescents is important in the context of escalating obesity rates and health issues related to poor diet and evidence that eating behaviors track across the lifespan and are influenced by nutrition knowledge and food literacy.

Two-dimensional food models appear to have potential as a teaching tool for nutrition education in the school environment as they offer visual and simulated hands-on experiences with food. They also have an advantage in terms of cost, weight, and size (storability and variety of foods) when compared to three-dimensional models but may be less durable.

Future work needs to look at the relationship between measures of sugar literacy and label reading behaviors and actual sugar consumption. Also, studies on brief nutrition education interventions using food models should involve a longer-term follow-up and more sensitive measurement tools.

## Figures and Tables

**Figure 1 nutrients-11-01763-f001:**
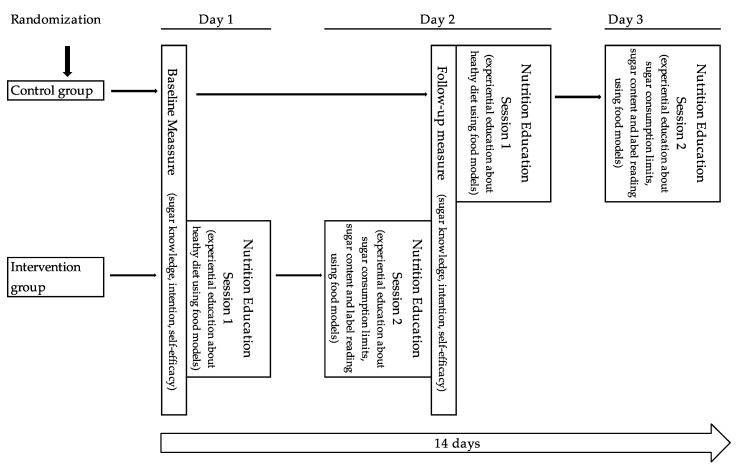
Research design diagram and timeline.

**Figure 2 nutrients-11-01763-f002:**
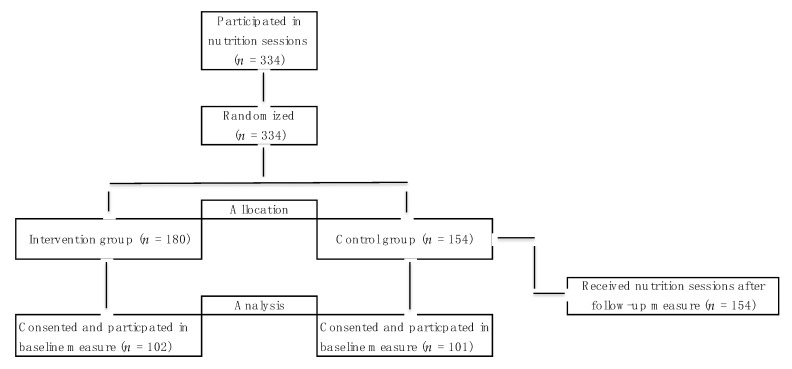
Flow of participants throughout the study.

**Figure 3 nutrients-11-01763-f003:**
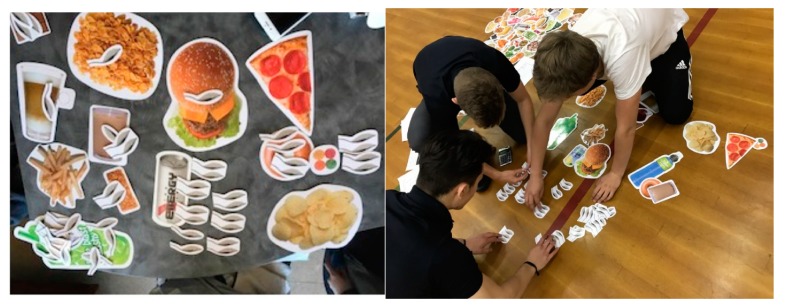
Pictures of interactive activity with two-dimensional food models.

**Table 1 nutrients-11-01763-t001:** Characteristics of participants at baseline.

Characteristic	Total (*n* = 203)		Control (*n* = 101)		Intervention (*n* = 102)		F (df)	*p*
	Mean (SD)	%	Mean (SD)	%	Mean (SD)	%		
Age (years)	15.9 (1)		15.4 (0.7)		16.4 (1.0)		70.34 (201)	0.00
Gender							16.29 (201)	0.00
Female		74.3		62.5	86.3			
Male		25.7		37.5	13.7			
Grade	10.0 (0.9)		9.5 (0.6)	0.6	10.4 (0.9)		72.72 (201)	0.00
9		31.1		51.0		10.8		
10		49.0		47.1		51.0		
11		12.1				24.5		
12		7.8		1.9		13.7		
Knowledge (correct answers)								
Daily portions of Food Groups				18.3		23.0		
Suggested maximum amount of added sugar				11.0		2.0		
Added sugar content in food and beverage				14.8		18.5		
Total Knowledge				15.7		18.6	2.92 (201)	0.09
Self-efficacy (5—35 ^a^)	23.9 (5.8)		23.17 (5.1)		24.7 (6.3)		2.58 (201)	0.11
Intention to consume less added sugar (3—21 ^b^)	14.4 (4.4)		13.9 (4.6)		14.9 (4.1)		3.70 (201)	0.06
Ability to interpret sugar content in food labels (1—7 ^c^)	4.8 (1.5)		4.9 (1.6)		4.6 (1.4)		1.81 (201)	0.18
Frequency of limiting foods high in added sugar (1—7 ^c^)	4.1 (1.5)		4.1 (1.6)		4.1 (1.4)		0.02 (201)	0.89
Frequency of added sugar being read in food labels (1—7 ^c^)	4.5 (1.9)		4.6 (1.9)		4.5 (1.8)		0.13 (201)	0.72

Note. SD = Standard deviation; F = F value; df = degrees of freedom; *p* = probability value. ^a^ Sum of five 7 point Likert scale questions. ^b^ Sum of three 7 point Likert scale question. ^c^ 7 point Likert scale

**Table 2 nutrients-11-01763-t002:** Means and repeated measures ANOVA for change pre–post intervention in knowledge, self-efficacy, intentions, ability to read and interpret food labels, and frequency of limiting food with added sugar.

Dependent Variable	Control (*n* = 101)		Intervention. (*n* = 102)		F (df)	*p*	η2
	Mean (SD)	% (SD)	Mean (SD)	% (SD)			
Knowledge (correct answers)							
Daily portions of Food Groups							
Baseline		18.3 (20.0)		23.0 (22.2)	59.84 (201)	0.00	0.23
Follow-up		21.0 (19.3)		59.3 (32.6)			
Suggested maximum amount of added sugar							
Baseline		11.0 (31.3)		2.0 (13.9)	138.14 (201)	0.00	0.41
Follow-up		14.0 (34.7)		75.0 (43.2)			
Added sugar content in food and beverage							
Baseline		14.8 (14.3)		18.5 (16.4)	64.68 (201)	0.00	0.24
Follow-up		17.7 (18.2)		47.7 (26.2)			
Total knowledge							
Baseline		15.7 (11.4)		18.6 (11.7)	104.84 (201)	0.00	0.34
Follow-up		20.1 (18.3)		54.5 (22.4)			
Self-efficacy (5—35 ^a^)							
Baseline	23.2 (5.1)		24.7 (6.3)		0.49 (201)	0.49	0.00
Follow-up	23.8 (5.8)		25.5 (6.0)				
Intention to consume less added sugar (3—21 ^b^)							
Baseline	13.9 (4.6)		14.9 (4.1)		4.93 (201)	0.03	0.02
Follow-up	13.3 (4.6)		15.3 (4.1)				
Ability to interpret sugar content in food labels (1—7 ^c^)							
Baseline	4.9 (1.6)		4.6 (1.4)		14.94 (200)	0.00	0.07
Follow-up	4.6 (1.7)		5.3 (1.8)				
Frequency of limiting foods high in added sugar (1—7 ^c^)							
Baseline	4.1 (1.6)		4.1 (1.4)		0.19 (201)	0.67	0.00
Follow-up	4.3 (1.7)		4.3 (1.6)				
Frequency of added sugar being read in food labels (1—7 ^c^)							
Baseline	4.6 (1.9)		4.5 (1.8)		3.42 (201)	0.07	0.02
Follow-up	4.5 (1.9)		4.8 (1.6)				

Note. SD = Standard deviation; F = F value; df = degrees of freedom; *p* = probability value; η2 = partial eta squared; ^a^ Sum of five 7 point Likert scale questions. ^b^ Sum of three 7 point Likert scale question. ^c^ 7 point Likert scale question.

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
