# Peer review of "Using Food Models to Enhance Sugar Literacy among Older Adolescents: Evaluation of a Brief Experiential Nutrition Education Intervention"

_nutrients, 2019, doi:10.3390/nu11081763_

Round 1

Reviewer 1 Report

The authors report results from a brief school-based nutrition lessons using life-sized 2-dimensional food models on adolescents’ sugar literacy

Introduction: 

It would be helpful to condense and focus the introduction (and discussion – to a lesser extent) on the objective of the intervention.  The extensive detail on behavior change theory does not seem particularly salient to objective/intervention.

Materials and Methods:

Research design paragraph – would be helpful to describe usual practice as “control group” and Intervention condition as “intervention group”.

Figure 1 – it would be helpful to provide a timeline to know the amount of time between baseline/follow-up measures and between the two nutrition sessions.  Also, in the figure, the follow-up measure appears to occur after the Nutrition sessions, which is misleading. 

How was randomization performed?  In the limitations, it suggests that randomization was performed by class, but also states that an intervention and control were included for each school? Please describe in the methods.

If individual students did not provide consent, how was this handled, as the nutrition sessions were provided to classes?

Figure 2 – this figure references “assessment for eligibility”, but no eligibility criteria were described in the methods.  Also, the information and numbers provided in the figure are very confusing with respect to the #’s who consented and those who were analyzed – Could this be simplified to reflect the most meaningful points relevant to the individuals not included in analyses?  Also, per the figure, it appears that all 154 individuals who “did not receive allocated intervention” (these are controls?) received the nutrition program after follow-up measure, even though 50 did not provide consent? 

Experimental variables – was the self-efficacy scale range 7 to 35 (as stated in methods) or 5 to 35 (as indicated in tables)?

Tables 1 & 2:

It would be helpful if the authors labeled the values in parentheses – some seem to be minimum/maximum of scale response options, though others are based on participants responses to open-ended questions.  For the knowledge questions – perhaps it would be more relevant and meaningful to present these data as the proportion of participants who responded correctly?

The authors should label or add footnotes to explain each of the columns in their tables.

The use of “change over time” might be better phrased as change pre – post intervention, as there are only two time-points involved.

Discussion:

The stated purpose of the study was “... to explore the impact of two 45-minute nutrition lessons using life-sized 2-dimensional food models, on adolescents’ sugar literacy”. The authors might discuss why sugar literacy information was only presented in one session, and one session covered Canada’s Food Guidelines.

Limitations: 

The authors might consider the timing of their follow-up assessment a limitation, as participants had not yet been provided an opportunity to implement what they had been instructed on. 

Reviewer 2 Report

·       In Fig.1 if the authors mention the type of intervention and parameters measured that will provide overall idea.

·       Did the authors perform gender based analysis

·       Authors are suggested to upload Appendix material correctly

·       Measurements performed only with questionnaire. Performing basic blood parameters might be more beneficial in addition to the classes and questionnaire
